# Pike: OTU-Level Analysis for Oxford Nanopore Amplicon Metagenomics

**DOI:** 10.3390/ijms26094168

**Published:** 2025-04-28

**Authors:** Danil V. Krivonos, Dmitry E. Fedorov, Dmitry N. Konanov, Andrey V. Vvedensky, Ignat V. Sonets, Elena V. Korneenko, Anna S. Speranskaya, Elena N. Ilina

**Affiliations:** 1Research Institute for Systems Biology and Medicine (RISBM), 18, Nauchniy Proezd, 117246 Moscow, Russiaignatsonets@gmail.com (I.V.S.); ilinaen@sysbiomed.ru (E.N.I.); 2Department of Molecular and Translational Medicine, Moscow Institute of Physics and Technology, State University, 141700 Dolgoprudny, Russia; 3Institute of Gene Biology Russian Academy of Sciences, 119334 Moscow, Russia; 4Saint Petersburg Pasteur Institute, Federal Service on Consumer Rights Protection and Human Well-Being Surveillance, 14, Mira Street, 197101 Saint Petersburg, Russia

**Keywords:** bioinformatics, metagenomics, sequencing

## Abstract

The Oxford Nanopore platform and nanopore sequencing are gaining increasing popularity in modern metagenomic research. However, there is a limited set of dedicated tools for analyzing this type of data. The tools used for nanopore amplicon sequencing data analysis often provide only taxonomy annotation without OTU sequence assembly. Conversely, tools that facilitate OTU assembly are constrained in their analysis to long reads, such as the V1–V9 regions of 16S rRNA for bacterial community studies or the full-length ITS cluster (ITS1–5.8S–ITS2) for fungal community studies. In other cases, researchers propose their own solutions without dedicated tools. In this paper, we present Pike, a novel tool for analyzing Oxford Nanopore amplicon sequencing data. Pike allows analysis without amplicon size limitations and allows de novo assembly of OTU sequences. In our research, we created mock communities of fungi and bacteria, which we then used to demonstrate the efficiency of our algorithm. Furthermore, we validated the algorithm using externally available data. We also compared our approach with similar ones to show its applicability.

## 1. Introduction

The majority of recent metagenomic studies have been conducted based on the Illumina platform. Furthermore, in general, the size of the fragment under investigation should not exceed the length of the paired Illumina reads. In this case, the selection of the fragment of interest is significantly limited, which in turn restricts the accuracy of the taxonomic classification of the studied community. At the same time, the Illumina reads are distinguished by their high quality, which has led to the advent of many bioinformatics tools that use this property [1,2,3]. Therefore, a standardized methodology has been designed for the analysis of metagenomic data obtained from platforms with high-quality reads. One of the most widely used and well-known software solutions in the field of metagenomics is the DADA2 package [1], which is based on the assumption that there will be a higher number of biologically correct sequences than those resulting from sequencing errors. Despite the significant advantages of Illumina, it still does not allow long amplicon sequences to be obtained.

The Oxford Nanopore platform is gaining popularity in the scientific community, enabling long reads from relatively short sample preparation times. Using nanopore sequencing, it becomes possible to completely close the variable regions such as V1–V9 of 16S rRNA and the ITS1–5.8S–ITS2 region. The analysis of such fragments will lead to an improvement in the accuracy of the taxonomic identification of bacterial and fungal communities. The interest in using nanopore sequencing in metagenomic studies has been visible for a long time in the literature [4,5,6,7,8]. In addition, an important advantage of the Oxford Nanopore platform is that flow cells can be used multiple times, eliminating the need to fully load the instrument, as with the Illumina platforms.

It should be noted that in a number of studies, researchers resort to creating their own individual approaches for nanopore amplicon data analysis. For instance, the authors of Ref. [9] provide the example of the identification of pure fungal isolates and fungal communities using three approaches. Other examples include raw nanopore reads and performing read correction and assembly using Canu [10], clustering sequences using VSEARCH [11], and clustering sequences using a modified version of NanoCLUST [12]. Approaches based on nanopore read clustering followed by consensus building were also successfully used in other early works [13,14]. For example, in the work of [14], focused on the analysis of fungal communities, read clustering was carried out using identity cutoffs using VSEARCH and subsequent consensus polishing using Medaka [15]. In other studies, reads obtained as a result of nanopore sequencing were simply mapped to a reference database [5]. Thus, approaches for analyzing amplicon data using clustering and read construction are already being used in practice by various groups. However, there are still very few ready-made tools for solving this problem, which further confirms the relevance of this task.

Among the existing tools, the most accessible method for analyzing amplicon sequencing data obtained using nanopore sequencing is the wf-metagenomics package from epi2me-labs [16]. Today, wf-metagenomics includes two algorithms: kraken2 [17] and minimap2 [18]. However, kraken2 classification could provide false positive classification [19]. Additionally, both approaches are also characterized by exclusively reference-based analysis, which does not allow the analysis of collected amplicon variants in the form of sequences of operational taxonomic units (OTUs).

In 2019, NanoCLUST was published as a tool for analyzing nanopore sequencing data based on clustering and subsequent consensus construction [12]. The NanoCLUST algorithm is based on the clustering of reads by their k-mer composition, using uniform manifold approximation and projection (UMAP), and the building of consensuses for each cluster using Canu [10], Racoon [20], and Medaka [15]. In the NanoCLUST article, the developers show its applicability only to the V1–V9 region of 16S rRNA. The NanoCLUST algorithm is designed to deal only with long amplicons, the size of which is greater than 1000 base pairs. This limitation comes from the Canu assembler, which is used on the NanoCLUST algorithm, therefore, the ability to analyze smaller amplicons is limited. This is especially critical in the competitive study of fungal communities, where the length distribution of full-length ITS clusters (ITS1-5.8S-ITS2) can range from 400 to 1000 bp.

Based on the idea of clustering reads in accordance with the composition of their k-mer signature using a parametric UMAP algorithm, we have developed our own tool with the aim of achieving high sensitivity and the generation of high-quality consensus sequences of any length. In addition to this feature, Pike has two modes of analysis: the single mode (where clustering and consensus building are performed independently for all samples) and the pool mode (where clustering and consensus building are performed jointly for all samples). The algorithm was implemented as a command-line tool, which we designated Pike.

## 2. Results and Discussion

### 2.1. Raw Data Processing

Sequencing resulted in a data set with an average depth of coverage of 180,861 reads per sample (V3–V4 dataset: 496277, V1–V9 dataset: 51908, ITS1 dataset: 180198, ITS2 dataset: 100285, ITS1-5.8S-ITS2 dataset: 75639). The quality of the resulting dataset was assessed using FastQC [21] and multiQC [22]. Read trimming was carried out using guppy_barcoding.

### 2.2. Mock Community Analysis

The results of the Pike analysis are presented in Figure 1, which visualizes the results of the analysis of bacterial and fungal mock communities. It was anticipated that each mock mixture would have an equal representation of each species adjusted for differences in sequence lengths. In the analysis of the mock mixtures, the taxonomy was determined using blastn with the criteria query coverage ≥60% and identity ≥95%. These criteria were chosen intentionally to ensure the accuracy and transparency of the results, including any redundancies, obtained using Pike.

For bacterial communities, when taxonomy is resolved to the genus level, the results are promising, with a general trend towards equilibrium and convergence across different replicates. It should be noted that communities that have undergone an additional amplification stage show some bias. This phenomenon is most pronounced in samples of long amplicons V1–V9. At the species level of bacterial community taxonomy, ambiguities emerge that are explained by the approach to determining taxonomy and the biological similarity between *Streptococcus salivarius* and *Streptococcus thermophilus*, *Klebsiella pneumoniae* and *Klebsiella variicola*, *Yersinia pseudotuberculosis* and *Yersinia pestis*, *Escherichia coli* and *Shigella boydii*. Additionally, it is observed that the shorter V3–V4 amplicons exhibit significantly reduced variation between replicates, which may be attributed to the higher accumulation of errors in long reads. A comparable phenomenon is also evident at the OTU level, where the results for V3–V4 closely mirror those obtained at the species level, whereas for V1–V9, there is considerable variability between different samples. On the other hand, the markedly superior outcomes for V3–V4 can be attributed to the deeper sequencing depth of the V3–V4 samples.

The results obtained for fungi at the genus level are highly consistent between replicates and between different types of amplicons. However, due to the presence of multiple fungal species within the same genus, the mock communities did not exhibit equal proportions. When the taxonomy is resolved to the species level, a fairly clear trend towards equal proportions is observed, with a general convergence between different types of amplicons with the exception of certain features, which we will discuss below. At the species level, the ITS1-5.8S-ITS2 amplicon also reveals a notable shift in fungal composition following reamplification. In particular, *Kluyveromyces marxianus* shows a strong drop in relative abundance, while *Yarrowia lipolytica* shows a comparative increase. The results obtained at the OTU level are consistent with the results obtained for bacterial compositions, where strong OTU divergence was also observed between samples with long reads.

Furthermore, it should be noted that the taxonomy of the fungus *Clavispora lusitaniae* remains unresolved in the ITS1 array. A number of *Clavispora lusitaniae* sequences were classified as simply *Clavispora* sp. However, all ten expected fungal species were identified. In the results for ITS2, *Meyerozyma guilliermondii* was identified as *Meyerozyma smithsonii*. However, according to the current taxonomic classification, *Meyerozyma smithsonii* is part of the *Meyerozyma guilliermondii* species complex [23]. Therefore, the results obtained are accurate.

To enable more reliable comparisons at the OTU level and to facilitate the analysis of samples with a limited number of reads, we developed a pool mode. As was demonstrated previously, the analysis of long amplicons of the V1–V9 type of the variable regions of 16S rRNA and full-length regions of ITS1-5.8-ITS2 presents certain challenges. Accordingly, we propose the use of pool mode as a solution in such cases. The results obtained by Pike with a pool mode are illustrated in Figure 2.

The analysis of an array of long amplicons using the pool mode provided the convergence of OTU sequences between samples. In the case of bacterial mock communities, the results were almost ideal, correlating with the expected composition of the bacterial mixture and repeating the previously observed amplification shift. In the case of fungal communities, the results were also well reproducible between different samples. Equal proportions of mixtures were also well reproduced, but in this case the number of OTUs did not correspond to the number of unique species, which could indicate some heterogeneity in the culture of the fungi available for analysis or technical Pike limitations.

It is important to note that, in practice, there are cases where multiple amplicon variants within the same species of microorganism are present in varying ratios. Thus, one variant may be considerably less abundant than the other. In such a case, some collisions may occur in Pike’s pool mode as a result of the merging of these OTU sequences into a single cluster. Also, a number of sequences may be clustered according to a common pattern of errors, and such cases cannot be easily caught. It is therefore recommended that the pool mode be used for relatively low numbers of data or for the detection of low-represented minor taxa, for which the number of reads in individual samples is insufficient to form a cluster for further detection. The results of the processing of sequencing data for other amplicons are also illustrated in Appendix A.

One of the indisputable advantages of the nanopore sequencing of amplicons is the absence of an additional instrumental stage of amplification, as in Illumina. To estimate the extent of the contribution of amplification, we conducted a reamplification of the initial mock communities and conducted a more detailed analysis in Appendix A.

### 2.3. Analysis of External Data

In this study, we did not intend to build ideal mock communities. Using microorganisms mixed in equal proportions, we did not expect to reflect the real biological communities. In practice, various approaches to metagenomic sequencing are validated through the analysis of commercially ready-made solutions, which feature combinations of microorganisms that can differ significantly, thus approaching natural microbial compositions. In particular, studies frequently use ZymoBIOMICS sequencing [24,25,26].

To further validate our approach, we conducted an external validation using a sequencing dataset of the popular microbial community ZymoBIOMICS Gut Microbiome Standard (Zymo Research, Irvine, CA, USA), which includes 14 bacterial species. The V1–V9 16S rRNA region sequencing dataset from the article referenced in [27] was taken. In this work, the authors provided the name of the used barcoding expansion kit, which allowed the removal of barcoding sequences through guppy_barcoding. Additionally, for data analysis using Pike, the primer sequences specified by the authors in the aforementioned publication were 27f and 1492r primers for 16S rRNA. The taxonomy for this analysis was performed with the criteria of blastn query coverage ≥98% and identity ≥97%. These criteria were chosen to reduce the underlying noise in relatively poorly classified sequences. The ZymoBIOMICS Gut Microbiome Standard composition is somewhat more complex than the one previously used. Therefore, to provide a clearer visualization of the data, more stringent criteria were employed for determining the taxon. The results are presented in Figure 3.

Additionally, the results are presented in Table 1. It was observed that the sequences designated as *g_Lachnospiraceae* UCG-008 and the *g_[Ruminococcus] gnavus* group, along with *Clostridioides difficile*, were incorrectly classified using the blastn + SILVAv138 databases. As a result, these sequences were excluded from the subsequent table, as they were not anticipated to be present in the bacterial composition. Additionally, minor members of the bacterial community, including *Salmonella enterica, Clostridium perfringens*, and *Enterococcus faecalis*, were not identified. The authors of the original publication [27] obtained comparable results when using the Illumina platform. These outcomes are likely attributable to the insufficient representation of these bacteria in the sample. Additionally, the authors of the original work observed the presence of *Salmonella enterica* in the samples obtained on the Oxford platform, which were not detected using Pike.

To validate the accuracy of the results obtained using Pike, we conducted an additional translation of the fastq file after trimming into fasta and performed a blast alignment of the sequences of 16S rRNA *Salmonella enterica* (NR_041696.1), 16S rRNA *Enterococcus faecalis* (NR_040789.1) and 16S *Clostridium perfringens* rRNA (NR_121697.2). In the cases of *E. faecalis* and *C. perfringens*, we did not identify any sequences with an identity to external sequences greater than 90%, while in the case of *S. enterica*, the situation was more ambiguous. This is because the identity of *S. enterica* and *E. coli* is greater than 97%, which generally makes this kind of assessment difficult. Probably, the results obtained by the authors were also a false positive inference due to the high identity of *S. enterica* and *E. coli*. Nevertheless, despite some ambiguities, it was noted that Pike returned results that were close to the expected bacterial composition.

A mock community does not fully reflect the real ratios of microorganisms, so we additionally demonstrated the work of our tool on external sequencing data of complete 16SrRNA amplicons from nasopharyngeal swab samples. The results of this analysis are presented in Appendix A.

### 2.4. Benchmarking and Comparison with Alternative Tools

A highly detailed and clear validation of the tool was attempted; however, due to the large volume of described material and graphic support, the validation and benchmarking were relocated to Appendix A and some figures were moved to Appendix A.

The existing approaches are limited in certain respects. Accordingly, we have proposed an alternative approach for analyzing amplicon sequencing data on the Oxford Nanopore platform. As we wrote earlier, among the methods for analyzing amplicon sequencing data obtained on the Oxford Nanopore platform, the most relevant approaches are epi2me-labs/wf-metagenomics [16] (including kraken2 [17] and minimap2 [18]) and NanoCLUST [12] (Appendix A). A comparative analysis of different tools was conducted on the mock community dataset that was previously created. The comparison results indicated that wf-metagenomics demonstrated a slight distortion of the original composition in both processing modes. Furthermore, the wf-metagenomics kraken2 mode with the NCBI_16s_18s_28s_ITS database was observed to have a tendency to confuse *Y. pseudotuberculosis* with *S. enterica*, which could potentially lead to an erroneous interpretation of the results. Additionally, as previously stated, wf-metagenomics lacks the capability to assemble OTU. Concurrently, NanoCLUST facilitated the examination of specific subsets, encompassing only the long amplicons V1-V9 16S rRNA and ITS1-5.8S-ITS2. This was due to an inherent technical limitation in the Canu algorithm, which has a minimum read size of 1000 nucleotide pairs. The analysis results are highly effective for V1-V9 at different taxonomic levels, with the recording of all eight target bacteria. However, NanoCLUST was found to be unable to detect all 10 target fungi for the ITS1-5.8S-ITS2 region, as the ITS1-5.8S-ITS2 sequences of some fungi are shorter than 400 bp. Therefore, *Clavispora lusitaniae*, *Candida auris*, and *Yarrowia lipolytica* were not identified by NanoCLUST. However, NanoCLUST proved to be an effective tool for analyzing bacterial communities using long amplicons. The results obtained via Pike provided a comprehensive analysis of the entire dataset studied, demonstrating the suitability and versatility of the proposed approach.

### 2.5. Discussion Summary

The objective of this study was to develop a tool for the analysis of amplicon sequencing data obtained on the Oxford Nanopore platform. The result was a command line tool we called Pike. In the course of this study, we also obtained mock communities that were mixed in equal proportions. We obtained such compositions solely for the purpose of convenient validation of our tool. These compositions do not fully reflect real natural microbial communities. However, it is important to emphasize that the creation of an optimal microbial mock community mixture was not the main objective of this study.

Based on the analysis of the obtained data, the effectiveness of the tool in restoring the expected microbial composition at different taxonomic levels with high accuracy was demonstrated. However, when executing the Pike algorithm in single mode, some divergence between samples was observed at the individual OTU level. At the same time, Pike with pool mode allows a significantly better study of metagenomic samples to be made at the OTU level, which makes it easier to compare different samples at the OTU level.

The mock communities obtained during this work do not accurately reflect the true compositions of natural microbial communities, where the relative abundance of organisms can vary significantly. To demonstrate the applicability of Pike to communities with unequal contents of various microorganisms, we utilized the ZymoBIOMICS Gut Microbiome Standard. According to the results of the mixture analysis using Pike, we successfully detected 11 out of 14 bacteria. Although three other bacteria, comprising less than 0.009% of the total microbial composition, were identified, it is important to note that these sequences were not detected during a manual search in the array of reads, suggesting their probable absence in the mixture. The results obtained show the efficacy of Pike in analyzing more complex communities.

In the wake of a comparative analysis with existing tools, it was shown that Pike is a universal tool that enables the analysis of microbial communities regardless of the length of the amplicon sequence.

## 3. Materials and Methods

### 3.1. Experiment Design

To validate the approach, artificial mixtures of various amplicons from bacterial and fungal mock communities were prepared. The bacterial mock communities included eight different bacterial species, while the fungal mock communities included ten different fungal species. For the bacterial communities, 16S rRNA amplicons V3–V4 and V1–V9 were obtained separately. For the fungal communities, ITS1, ITS2, and the full-length ITS cluster (ITS1-5.8S-ITS2) were used. Then, for each type of amplicon, they were mixed in equal proportions. Such mixtures were prepared independently in triplicate.

Furthermore, in order to assess the impact of amplification on the alteration of the initial microbial composition, all the resulting mixtures were separated into two aliquots, one of which was subjected to an additional phase of amplification. Table 2 illustrates the composition of the mock communities.

#### 3.1.1. Amplification of 16S rRNA

The 16S V3–V4 region was amplified using primers 341F and 785R [28]. The 16S V1–V9 region was amplified with primers 27F and 1492R [29]. The primer sequences are shown in Table 3. PCR was performed in a total volume of 25 μL, containing 1X PCR-mix blue (Amplisens), 0.2 mM dNTP, 400 nM of each primer and 1–5 ng of bacterial DNA. The reaction was subjected to 95 °C for 2 min, followed by 25 cycles of 95 °C for 15 s, 55 °C for 25 s, 72 °C for 40 s (V3–V4) or 90 s (V1–V9), and final elongation 72 °C for 5 min in a C1000 Touch Thermal Cycler (Bio-Rad, Hercules, CA, USA). 

#### 3.1.2. Amplification of Fungal ITS1, ITS2 Region

Amplification of the ITS1-5.8S-ITS2 region was carried out using primers ITS5 and ITS4. ITS1 region was amplified with primers ITS5 and ITS2. Primers ITS3 and ITS4 were used for ITS2 region amplification [30]. The primer sequences are shown in Table 3. PCR reaction protocol included 1X PCR-mix blue (Amplisens), 0.2 mM dNTP, 400 nM of each primer and 1–5 ng of fungal DNA in a 25 μL.

The thermal cycling conditions for polymerase chain reaction (PCR) were as follows: an initial denaturation at 95 °C for 2 min, 30 cycles of 95 °C for 15 s, 54 °C for 25 s, 72 °C for 60 s, and then a final extension at 72 °C for 5 min, conducted in a C1000 Touch Thermal Cycler (Bio-Rad).

### 3.2. Amplicon Compositions and Reamplification

PCR products were purified using magnetic beads (MGI). Amplicons with the same target were mixed in equal masses. Three microliters of each mixture were used for the subsequent round of PCR (10 cycles) with the same primers in the same conditions previously described, and purified using magnetic beads. The mixing of amplicons and reamplification was performed in triplicate.

#### Library Preparation and Sequencing

Library preparation of PCR products and reamplified PCR products was conducted in triplicate using the Ligation Sequencing Kit (SQK-LSK109) and the Native Barcoding Expansion Kit (EXP-NBD104 and EXP-NBD114), in accordance with the Native Barcoding Amplicon protocol.

High-throughput sequencing was conducted on the PromethION instrument (Oxford Nanopore Technologies, Oxford, UK). The basecalling was performed using Guppy v5.1.13.

### 3.3. Algorithm Explanation

The Pike algorithm is based on four fundamental stages: data preparation, feature collection, clustering, and consensus building. Figure 4 illustrates the schematic diagram of the algorithm.

#### 3.3.1. Reads Preparation

The initial stage of the pipeline is the data processing stage. In the preliminary stage, barcoding sequences are removed from the reads. In the course of our research, we used the guppy_barcoding pipeline for the removal of barcoding sequences. It should be noted that the barcoding sequences are not automatically removed by the tool. It is therefore recommended that existing tools for trimming barcoding sequences be used. The presence of barcoding sequences would have a significant impact on subsequent analysis, as they would result in the generation of redundant consensus sequences. Additionally, it has been observed that approaches that cut adapters with automatic detection, such as porechop, are not optimal for analyzing 16S data with our tool. This may lead to certain issues, which have been further described in Appendix A.

After that, the reads where either F or R primer is not presented are removed via cutadapt and are not used in the further analysis [31]. Furthermore, following this procedure, all reads are oriented in a single direction, with consideration given to complementarity. Following this, reads are filtered according to the Q20+ requirements, based on the median quality of the reads. Additionally, the reads are filtered based on their length, ensuring that the minimum and maximum expected lengths are met. In our analysis, the following criteria were selected and are presented in Table 4.

Next, the quality of the reads is additionally evaluated manually. It was observed that a number of positions exhibited a quality significantly higher than the median quality (Q90). This suggests the presence of multiple such positions. This is likely attributed to the fact that Q90s are reduced to FP16 numerical precision, which causes us to roll over 1.0 and limit Qinf to 90. As a result, the positions with Q90 quality are excluded from the calculation of median quality.

#### 3.3.2. Feature Collection

The next step is the collection of features. In this step, the k-mer signature is collected by pre-squeezing the homopolymers according to the following procedure: …*XYYYYXXY... → …XYX…*. The decision to use compressed reads was motivated by the fact that the reads obtained via the Oxford Nanopore platform are of low quality in the region with homopolymers and are quite often mistaken in their quantity [32]. The presence of differences in homopolymers can result in the excessive clustering of reads. A k-mer signature is defined as the frequency of occurrence of each possible k-mer (by default, we chose k = 6). The user can adjust the value of k depending on the data. The validation of the k-measure size is given in Appendix A. Following this procedure, a vector of values is obtained for each read. A matrix with dimensions of N × 4^k^ is obtained for the whole sample, where N is the number of reads that went for analysis and k is the length of the k-mer. Compressed versions of reads are used only for feature collection and will not be used in building consensus. The collected feature matrix is subjected to central log ratio transformation in order to adequately handle the compositional data.

#### 3.3.3. Clustering

The matrix of features obtained at the preceding stage is subjected to a procedure of successive dimensionality reduction. At the initial stage, the matrix dimension is reduced using principal component analysis (PCA) to 30 components. Thus, the matrix obtained following the PCA dimensionality reduction process has dimensions N × 30. The preliminary use of PCA was necessary to speed up the calculation and suppress some of the noise level. Then, this matrix is subjected to the next stage of dimensionality reduction using the uniform manifold approximation and projection for dimension reduction (UMAP) algorithm, resulting in a matrix of dimensions N × 2. This results in a matrix with dimensions N × 2. Our tool uses parametric UMAP to minimize parameter optimization [33].

Further, the hierarchical density-based spatial clustering of applications with noise (HDBSCAN) algorithm, with the Euclidean distance metric, is applied to determine clusters. It is not optimal to include all reads within a cluster in the consensus building process. Therefore, clusters are «filtered» and only a subset of reads is selected. In this step, reads whose lengths fall outside the μ ± 10 range of the read length distribution are discarded. Similarly, reads that fall outside the μ ± 2σ distribution of the GC composition distribution are also excluded. The selected reads are then subjected to a multiple alignment procedure using MAFFT with default parameters [34]. MAFFT was used because of the speed of the multiple alignment procedure.

#### 3.3.4. Consensus Building

For each cluster, we positionally assemble a consensus sequence, where a possible option (A, T, G, C or gap) is evaluated in each position. In the event that the gap is the most common, the character “-” is entered. In all other cases, the letter with the highest average quality in the reads is selected. At the same time, the letter that is supported by at least 3 reads should correspond to the position with the highest average quality. In the case where the most frequently occurring letter has a quality lower than any less frequently occurring letter, N is placed. In instances where a position fails to meet the aforementioned criteria, a specific position is defined as N. The resulting consensus is called proto-consensus. All gaps (represented by «-» symbols) are then removed from the proto-consensus sequence. Finally, the polisher Medaka algorithm is applied [15] to obtain the final form of the consensus. The conventional visualization of the proto-consensus building schemes is presented in Figure 5.

Thus, a set of amplicon variants is obtained, where the number of reads related to the cluster will reflect the representation of each amplicon variant in the sample.

In the case of untrimmed reads, the majority of degenerate positions with the letter N are observed at the beginning and end of the assembled consensus. This is likely due to the fact that the quality at the end of the readings is extremely low. Therefore, a separate parameter has been provided for processing such reads, which trims the poly-N at the ends of the consensus sequences.

#### 3.3.5. Mode Explanation

The Pike algorithm comprises two distinct operational modes. The so-called single mode is in which the analysis of each sample is carried out independently. In this case, feature collection, clustering and consensus building are carried out independently for each sample. The second pool mode initially groups all data from all samples, performs joint clustering of all reads from the whole dataset, and builds consensus sequences based on all samples. The final stage of the pool mode is the restoration of the composition of consensus sequences for each individual sample.

Pool mode is recommended for cases of poor-quality data or a small amount of initial information that does not allow samples to be studied independently in a single mode. Additionally, pool mode is recommended for mainly long reads and for a number of short reads, as difficulties are observed when analyzing data at the OTU level.

#### 3.3.6. Taxonomy Identification

To taxonomically annotate the assembled OTU sequences, we performed a BLASTn search using the SILVAv138 database [35] for 16S sequences and the UNITE database (unite_04.04.2024) [36] for fungal ITS sequences. The motivation for choosing a taxonomy determination method was caused by the inaccuracy in the application of standard approaches like dada2 assign taxonomy for fungal sequences [37]. In this case, taxonomy identification using blastn was used solely for our convenience. Furthermore, we developed an additional Pike block as a command-line tool, get_taxonomy.py, to automate the process of taxonomic identification. For more accurate taxonomy analysis, we also recommend using external software that is specifically designed for this task. Such specialized tools include IDTAXA [38], MAPSeq [39], QIIME [40], SPINGO [41] and many other instruments.

## 4. Conclusions

The growing interest in Oxford Nanopore amplicon metagenomics within the scientific community can be primarily attributed to the capacity to analyze long amplicons, including those spanning the V1–V9 regions and the full-length ITS region (ITS1-5.8S-ITS2). In this work, we present a new tool for the analysis of any length amplicon sequencing data obtained on the Oxford Nanopore platform. We validated the tool using our own mock compositions of fungal and bacterial communities, benchmarked it, and described a more detailed analysis of mock communities. Furthermore, an external dataset was analyzed as well, which demonstrated the applicability of the tool to open data. Our tool was also validated and its approach was compared with that of alternative tools.

## Figures and Tables

**Figure 1 ijms-26-04168-f001:**
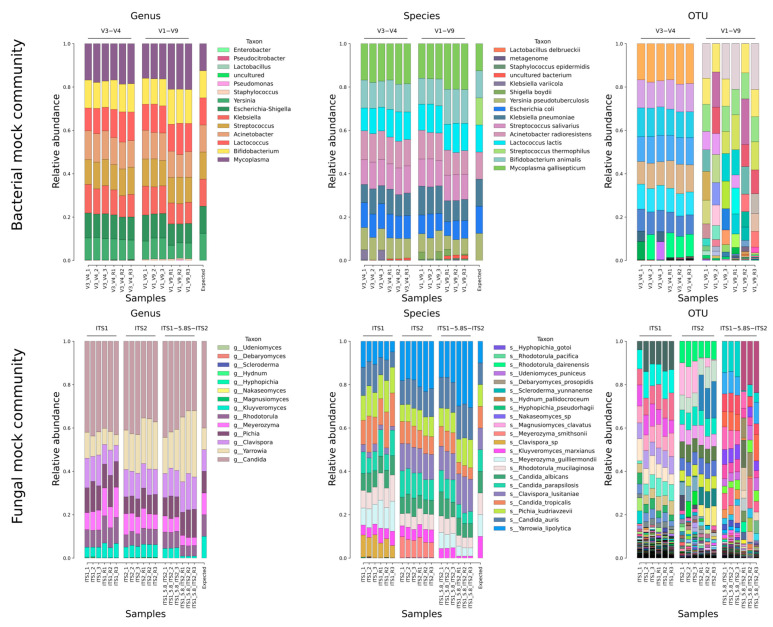
Pike mock community processing results on different taxonomy levels (results were obtained via Pike in the single mode). Upper figures—Pike’s results for bacterial mock community; Lower figures—Pike’s results for fungal mock community.

**Figure 2 ijms-26-04168-f002:**
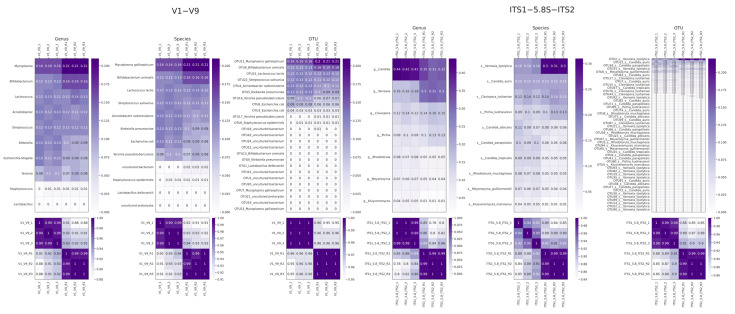
Pike mock community processing results with pool mode. Upper figures—results of assessing relative representation at different taxonomy levels. The lower pictures are the values of the Pearson correlation coefficient. Zero values in the picture correspond to values <0.001.

**Figure 3 ijms-26-04168-f003:**
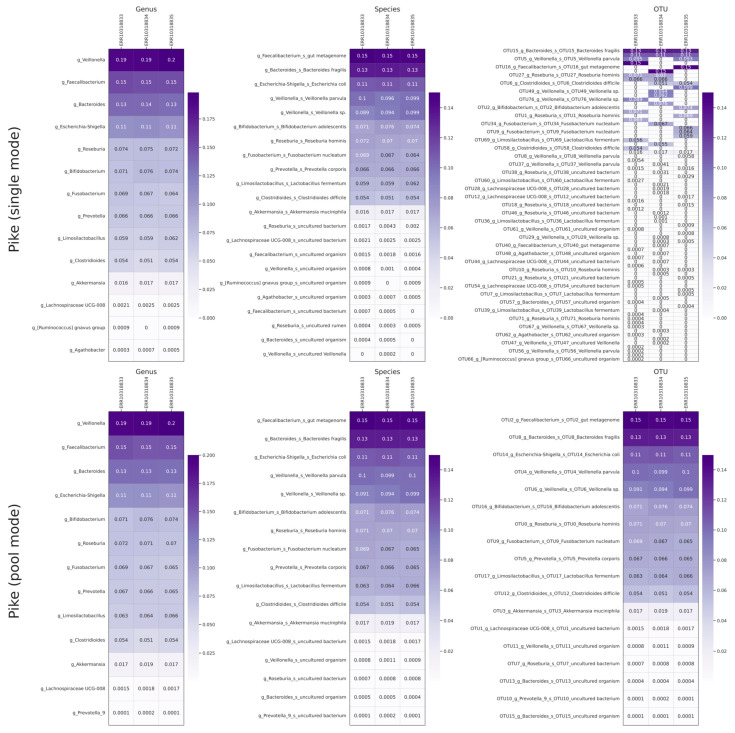
The results of the analysis of the external sequencing data of the synthetic microbial community ZymoBIOMICS Gut Microbiome Standard using Pike with single and pool modes (relative abundances are rounded to four decimal places).

**Figure 4 ijms-26-04168-f004:**
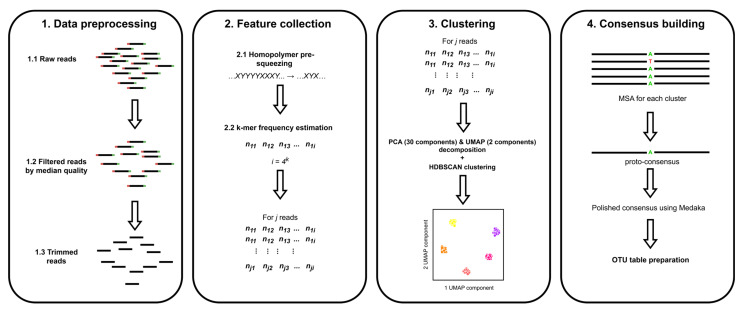
The schematic diagram illustrates Pike’s algorithm.

**Figure 5 ijms-26-04168-f005:**
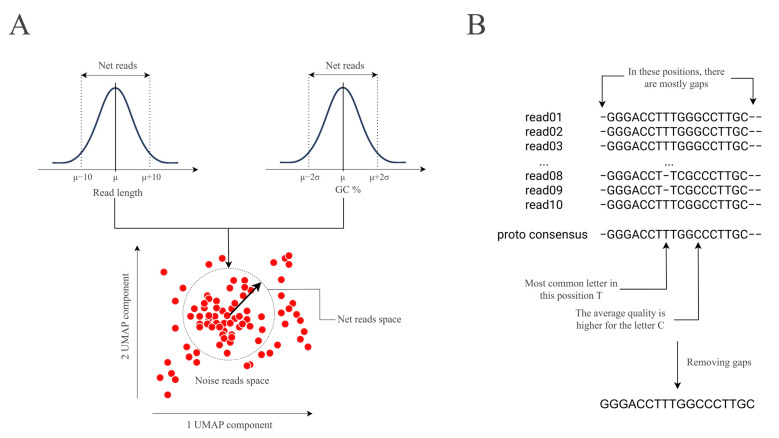
Proto-consensus building scheme. (**A**) Visualization of the selection of reads for consensus construction. (**B**) Scheme for selecting consensus letters from multiple alignments.

**Table 1 ijms-26-04168-t001:** Relative abundances for ZymoBIOMICS Gut Microbiome Standard obtained via Pike.

		Single Mode			Pool Mode		
Species	Expected	ERR10318833	ERR10318834	ERR10318835	ERR10318833	ERR10318834	ERR10318835
*Veillonella rogosae*	*0.1587*	0.190446	0.191516	0.1983	0.1927	0.194628	0.200128
*Faecalibacterium prausnitzii*	*0.1763*	0.152488	0.152488	0.1489	0.1498	0.1459	0.1466
*Bacteroides fragilis*	*0.0994*	0.1348	0.1350	0.1328	0.1344	0.1343	0.1328
*Escherichia coli*	*0.1212*	0.1090	0.1101	0.1066	0.1088	0.1098	0.1069
*Bifidobacterium adolescentis*	*0.0878*	0.0707	0.0760	0.0745	0.0708	0.0762	0.0742
*Roseburia hominis*	*0.0989*	0.0743	0.0749	0.0721	0.0718	0.0708	0.0703
*Fusobacterium nucleatum*	*0.0749*	0.0693	0.0669	0.0645	0.0692	0.0668	0.0646
*Prevotella corporis*	*0.0498*	0.0663	0.0658	0.0661	0.0667	0.0655	0.0654
*Lactobacillus fermentum*	*0.0963*	0.0590	0.0587	0.0620	0.0629	0.0643	0.0661
*Clostridioides difficile*	*0.0262*	0.0540	0.0510	0.0536	0.0543	0.0511	0.0539
*Akkermansia muciniphila*	*0.0970*	0.0164	0.0174	0.0167	0.0168	0.0187	0.0172
*Salmonella enterica*	*0.00009*	0	0	0	0	0	0
*Clostridium perfringens*	*0.000002*	0	0	0	0	0	0
*Enterococcus faecalis*	*0.000009*	0	0	0	0	0	0

**Table 2 ijms-26-04168-t002:** Mock community.

Fungal Composition	Bacterial Composition
*Clavispora lusitaniae* *Candida tropicalis* *Kluyveromyces marxianus* *Candida auris* *Pichia kudryavtseva* *Candida albicans* *Rhodotorula mucilaginosa* *Candida parapsilosis* *Meyerozyma guilliermondii* *Yarrowia lipolytica*	*Streptococcus thermophilus* *Lactococcus lactis* *Bifidobacterium animalis* *Escherichia coli* *Yersinia pseudotuberculosis* *Klebsiella pneumoniae* *Acinetobacter baumannii* *Mycoplasma gallisepticum*

**Table 3 ijms-26-04168-t003:** Primers used.

Mock Community Type	Amplicon Region	Forward Sequence (5′–3′)	Reverse Sequence (5′–3′)	Source
Bacteria	V3–V4	CCTACGGGNGGCWGCAG	GACTACHVGGGTATCTAATCC	[28]
	V1–V9	AGRGTTYGATYMTGGCTCAG	CGGYTACCTTGTTACGACTT	[29]
Fungi	ITS1	GGAAGTAAAAGTCGTAACAAGG	GCTGCGTTCTTCATCGATGC	
	ITS2	GCATCGATGAAGAACGCAGC	TCCTCCGCTTATTGATATGC	[30]
	ITS1-5.8S-ITS2	GGAAGTAAAAGTCGTAACAAGG	TCCTCCGCTTATTGATATGC	

**Table 4 ijms-26-04168-t004:** Expected length of parameters used.

	Bacteria		Fungi		
	16S V3–V4	16S V1–V9	ITS1	ITS2	ITS1-5.8S-ITS2
Minimum expected length (bp)	350	900	100	100	300
Maximum expected length (bp)	600	1600	500	500	1200

## Data Availability

All data obtained during the work was uploaded to NCBI BioProject under the accession number PRJNA1147009. Reads were uploaded with also trimmed ONT barcodes. Pike code is available via GitHub: https://github.com/DanilKrivonos/Pike (accessed on 20 April 2025) and it can be downloaded via pip (pip install pike-meta). Benchmarking code and visualization code are available via link: https://github.com/DanilKrivonos/Pike_work_notebooks (accessed on 20 April 2025). The demo dataset is available in Zenodo (https://doi.org/10.5281/zenodo.13730430).

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
