# Peer review of "Pike: OTU-Level Analysis for Oxford Nanopore Amplicon Metagenomics"

_ijms, 2025, doi:10.3390/ijms26094168_

Round 1
Reviewer 1 Report
Comments and Suggestions for Authors
- General
This manuscript presents a workflow that uses Nanopore long‑read sequencing for amplicon‑based OTU metagenomic analysis. Although the target community (researchers working specifically with Nanopore long‑amplicon data) is relatively small, the paper delivers a practical toolset that those researchers will welcome. The overall scientific quality, structure, and validation experiments merit publication once minor issues are addressed. - Good points
This study delivers a genuinely helpful workflow for Nanopore long‑amplicon data.
The authors offer functionality missing from the field by lifting the read‑length limits that hamper existing pipelines and adding a pool mode to stabilize OTU calls at low coverage.
The software is open‑source, pip‑installable, and well-documented, and the accompanying data make every analysis step reproducible.
Although the user community for Nanopore amplicon work is still relatively small, those researchers will gain an immediate, practical benefit from the tool, underscoring the paper’s relevance.
- Weak points
The main weakness is the lack of a real‑world dataset: all evaluations rely on mock or commercial standards. Adding even a single environmental or clinical sample would show that the pipeline performs well in truly complex communities.
The manuscript also contains a noticeable number of typographical and grammatical errors.
typo.- 2.1 Experement design -> 2.1 Experiment design
- guppi_barcoding -> guppy_barcoding (3 places)
- nootebook -> notebook ( l.465, the URL is accessible but "noote." is it OK?)
- There may be many more, please use the tool to check.
In several figures, especially Figures 3–5, labels are too small to read comfortably; enlarging fonts would improve clarity.
Addressing these points should be straightforward and warrant acceptance after minor revision.
- Recommendation
Requests in minor rivision.
The manuscript is publishable after the authors
(i) add a concise natural‑sample demonstration or justify its absence,
(ii) correct language errors, and
(iii) improve figure legibility.
I look forward to seeing the revised version online soon.
Author Response
Dear Reviewer,
Thank you for your comments and remarks on the work and the results obtained. We have taken care to respond to all of them and to reflect them in the text of the manuscript. We took your advice and ran Pike on the external fifty-seven nasal swabs sequencing data from the paper (https://doi.org/10.3390/genes11091105). We have fixed all typos, including correcting the URL and updating the link in the text. We have also increased the font size for images as much as possible.
Reviewer 2 Report
Comments and Suggestions for Authors
This paper introduces a novel bioinformatics tool named Pike for analyzing Oxford Nanopore amplicon sequencing data, with particular focus on microbial community studies targeting 16S rRNA and ITS regions. The tool's key innovation lies in its capability to support analysis of amplicons with arbitrary lengths while enabling de novo assembly of Operational Taxonomic Unit (OTU) sequences. The experimental design is well-conceived, employing both mock communities and publicly available datasets for validation, along with comparative analyses against existing tools such as NanoCLUST and wf-metagenomics. Overall, this study demonstrates significant scientific merit, though several aspects require improvement.
(1) The "Feature collection" section lacks sufficient detail regarding the specific implementation of k-mer compression (homopolymer squeezing). For instance, how were different k-values selected? Were tests conducted to evaluate the impact of varying k-values on clustering performance?
(2) The analysis of mock communities revealed relatively high OTU-level variability in V1-V9 long amplicons, yet the underlying reasons were not thoroughly discussed. Could this be attributed to characteristic error patterns in Nanopore sequencing, such as homopolymer errors?
(3) The comparative evaluation against NanoCLUST and wf-metagenomics was limited to mock communities, without testing on real-world samples (e.g., soil or gut microbiome datasets). Furthermore, quantitative comparisons of computational efficiency metrics, including runtime and memory usage, were absent.
This work presents a valuable new tool for the field, though certain methodological details, result interpretations, and discussion points need enhancement. The manuscript may be considered for acceptance after appropriate revisions.
Author Response
Dear Reviewer,
Thank you for your comments and remarks on the work and the results obtained. We have taken care to respond to all of them and to reflect them in the text of the manuscript.
(1) We tested the value of k. This was included in Supplementary 2 5. K-mer size validation. Based on the validation results, we decided to set the default value of k equals 6. This parameter can be changed depending on the data (argument -k). We added a link to the Supplementary to the text.
(2) We did not find any recognizable ONT error patterns or errors in homopolymer regions.
(3) We followed your advice and ran Pike on external sequencing data from fifty-seven nasal swabs from the paper (https://doi.org/10.3390/genes11091105). However, it is always challenge to compare different tools efficiency on real data without reference values. That is why we showed the comparison only on data with a known microbial composition. We also measured the execution time and memory usage and reported it in Supplementary 2 1.
Round 2
Reviewer 2 Report
Comments and Suggestions for Authors
The manuscript has been sufficiently improved to warrant publication in IJMS